

# lncRNA TMEM51-AS1 and RUSC1-AS1 function as ceRNAs for induction of laryngeal squamous cell carcinoma and prediction of prognosis

Lian Hui[1], Jing Wang[1], Jialiang Zhang[1] and Jin Long[2]

[1] Department of Otolaryngology, the First Hospital of China Medical University, Shenyang, Liaoning Province, China
[2] Department of General Surgery, the First Hospital of China Medical University, Shenyang, Liaoning Province, China

Corresponding author
Jin Long, jinlong20187@126.com

## ABSTRACT

**Background.** Long non-coding RNAs (lncRNAs) can function as competing endogenous RNAs (ceRNAs) to interact with miRNAs to regulate target genes and promote cancer initiation and progression. The expression of lncRNAs and miRNAs can be epigenetically regulated. The goal of this study was to construct an lncRNA-miRNA-mRNA ceRNA network in laryngeal squamous cell carcinoma (LSCC) and reveal their methylation patterns, which was not investigated previously.

**Methods.** Microarray datasets available from the Gene Expression Omnibus database were used to identify differentially expressed lncRNAs (DELs), miRNAs (DEMs), and genes (DEGs) between LSCC and controls, which were then overlapped with differentially methylated regions (DMRs). The ceRNA network was established by screening the interaction relationships between miRNAs and lncRNAs/mRNAs by corresponding databases. TCGA database was used to identify prognostic biomarkers.

**Results.** Five DELs (downregulated: TMEM51-AS1, SND1-IT1; upregulated: HCP5, RUSC1-AS1, LINC00324) and no DEMs were overlapped with the DMRs, but only a negative relationship occurred in the expression and methylation level of TMEM51-AS1. Five DELs could interact with 11 DEMs to regulate 242 DEGs, which was used to construct the ceRNA network, including TMEM51-AS1-miR-106b-SNX21/TRAPPC10, LINC00324/RUSC1-AS1-miR-16-SPRY4/MICAL2/ SLC39A14, RUSC1-AS1-miR-10-SCG5 and RUSC1-AS1-miR-7-ZFP1 ceRNAs axes. Univariate Cox regression analysis showed RUSC1-AS1 and SNX21 were associated with overall survival (OS); LINC00324, miR-7 and ZFP1 correlated with recurrence-free survival (RFS); miR-16, miR-10, SCG5, SPRY4, MICAL2 and SLC39A14 were both OS and RFS-related. Furthermore, TRAPPC10 and SLC39A14 were identified as independent OS prognostic factors by multivariate Cox regression analysis.

**Conclusion.** DNA methylation-mediated TMEM51-AS1 and non-methylation-mediated RUSC1-AS1 may function as ceRNAs for induction of LSCC. They and their ceRNA axis genes (particularly TMEM51-AS1-miR-106b-TRAPPC10; RUSC1-AS1-miR-16-SLC39A14) may be potentially important prognostic biomarkers for LSCC.

## INTRODUCTION

Laryngeal squamous cell carcinoma (LSCC) is one of the common malignancies of the upper respiratory tract that has been associated with a deterioration of the environment and an increase in the occupational stress. It was estimated that 13,360 new cases were diagnosed in 2017 in the United States, of which over 3,660 were fatal (*Siegel, Miller & Jemal, 2017*). In China, an estimated 26,400 new cases of LSCC and 14,500 cancer-related deaths also occurred in 2015 (*Chen et al., 2016*). Although patients with LSCC can be managed by surgical intervention, radiation therapy and chemotherapy, the overall five-year survival remains poor (approximately 60%) (*Rudolph et al., 2011*). Therefore, there is an urgent need to deeply understand the molecular mechanisms underlying LSCC carcinogenesis or progression in order to develop more effective therapeutic strategies.

Accumulating evidence has suggested that non-coding RNAs (ncRNAs) play crucial roles in the initiation and development of tumors. ncRNAs are loosely categorized into small ncRNAs and long non-coding RNAs (lncRNAs), both of which have regulatory functions in various biological processes. The well-documented small ncRNAs are microRNAs (miRNAs; ~22 nucleotides long) that regulate gene expression by binding to complementary sequences in the 3′ untranslated region (UTR), leading to either inhibition of translation or degradation of the transcripts (*Jean & Mihaela, 2014*). Although the mechanisms remain unclear, growing evidence supports that lncRNAs could function as competing endogenous RNAs (ceRNAs) by competitively binding to miRNAs through their miRNA response elements (MRE) and subsequently regulate target RNA expression (*Salmena et al., 2011*). This ceRNA mechanism has generated much interest to explain tumor development and progression in many malignancies, such as gastric cancer (*Song et al., 2018*), thyroid carcinoma (*Zhao et al., 2018*) and hepatoblastoma (*Liu et al., 2017a*). Recent studies also have preliminarily revealed several underlying ceRNA regulatory interactions in LSCC. Luciferase reporter assay and Western blotting results suggested that AC026166.2-001 could act as a sponge of miR-24-3p and regulate the expression of p27 and cyclin D1 (*Shen et al., 2018*). lncRNA H19 was shown to serve as a ceRNA by sponging miR-148a-3p to upregulate the target gene DNA methyltransferase 1 (*Wu et al., 2016*). NEAT1 was also reported to regulate the expression of cyclin dependent kinase 6 through modulating miR-107 (*Wang et al., 2016*). Furthermore, a ceRNA network, including 30 genes, 21 miRNAs and 19 lncRNAs was also built based on microarray analysis of 6-paired clinical samples in LSCC (*Zhang et al., 2016*). However, analysis of the lncRNA-miRNA-mRNA regulatory network of LSCC with larger sample sizes and confirmation of their clinical associations are still lacking.

In addition, DNA methylation has been identified as an important mechanism to regulate gene expression in cancer cells epigenetically, which not only regulates the expression of protein-encoding genes, but also affects miRNAs and lncRNAs. For example, hyper-methylation of the promoter region was observed to lead to a loss of expression of lncRNA maternally expressed gene 3 (MEG3). Downregulated MEG3 was insufficient to sponge miR-9 and block its inhibition effects on the expressions of E-cadherin and FOXO1, consequently resulting in poor prognosis in patients with esophageal squamous cell

carcinoma (*Dong et al., 2017*). The study of *Guo et al. (2018a)* also suggested lncRNA CTC-276P9.1 was hyper-methylated in esophageal squamous cell carcinoma. Over-expression of CTC-276P9.1 inhibited cancer cell proliferation and invasion *in vitro* probably by regulating epithelial-mesenchymal transition. *Liao et al. (2015)* identified 761 lncRNA genes with DNA hyper-methylation in colorectal cancer using a free MethylCap-seq dataset. *Cheung & Lee (2010)* found that the loci of three miRNAs (namely miR-199a-2, miR-124a-2 and miR-184) were linked to hyper-methylated differentially methylated regions (DMRs) in human testicular cancer. However, the DNA methylation regulatory mechanisms of miRNAs and lncRNAs have rarely been reported in LSCC.

The goal of this study was to establish an lncRNA-miRNA-mRNA ceRNA network in LSCC using larger samples and to investigate their methylation patterns. Our results may provide new clues for biologists to further understand the pathogenesis of LSCC.

## MATERIAL AND METHODS

### Data source

lncRNA, miRNA, mRNA and methylation data were retrieved from Gene Expression Omnibus (GEO) database (http://www.ncbi.nlm.nih.gov/geo/) in January 2018 according to the following inclusion criteria: (1) lncRNA, miRNA, mRNA expression or methylation profiles; (2) laryngeal tissue samples, not blood, interstitial fluid or cells; (3) inclusion of control; (4) human samples; and (5) patients with LSCC.

Two lncRNA microarray datasets were obtained under accession number GSE59652 (7 LSCC and 7 paired adjacent normal tissues) (*Shen et al., 2014*) and GSE84957 (9 LSCC and 9 paired adjacent non-neoplastic tissues) (*Feng et al., 2016*). The microarray platforms of GSE59652 and GSE84957 were Agilent-033010 (GPL13825, Arraystar Human LncRNA microarray V2.0) and Agilent-042818 (GPL17843, Agilent-042818 Human lncRNA Micorarray 8_24_v2), respectively.

Two miRNA microarray datasets were collected under accession number GSE70289 (12 LSCC tissues and 4 adjacent normal tissues) (*Karatas et al., 2015*) and GSE62819 (5 LSCC carcinoma and 5 paired adjacent non-neoplastic tissues). The microarray platforms of GSE70289 and GSE62819 were Agilent-031181 (GPL15018, Unrestricted_Human_miRNA_V16.0_Microarray 030840) and Affymetrix Multispecies miRNA-3 Array (GPL16384), respectively.

Four mRNA microarray datasets were available under accession number GSE51985 (10 LSCC and 10 paired adjacent normal tissues), GSE84957 (9 LSCC and 9 paired adjacent normal tissues) (*Feng et al., 2016*), GSE59102 (29 LSCC and 13 normal margin tissues) and GSE58911 (15 LSCC and 15 normal tissue distant to LSCC) (*Sharon et al., 2015*). The microarray platforms of GSE51985, GSE84957, GSE59102 and GSE58911 were Illumina HumanHT−12 V4.0 (GPL10558), Agilent-042818 (GPL17843, Human lncRNA Micorarray 8_24_v2), Agilent-014850 (GPL6480, Whole Human Genome Microarray 4x44K G4112F) and Affymetrix Human Gene 1.0 ST Array (GPL6244), respectively.

One set of DNA methylation data was acquired under accession number GSE25093 (*Poage et al., 2012*; *Poage et al., 2011*) which included 213 blood and 109 tissue samples.

Among the 109 tissue samples, 56 were isolated from oral, 16 from pharyngeal, and 22 from laryngeal origin, while 15 were of unclear origin. Thus, only these 22 samples from laryngeal origin (15 LSCC tissues and 7 controls) were used in our study. The microarray platform of GSE25093 was Illumina HumanMethylation27 BeadChip (GPL8490, HumanMethylation27_270596_v.1.2).

The mRNA and miRNA Seq-data of head and neck squamous cell carcinoma (Level 3) were also downloaded from The Cancer Genome Atlas (TCGA; https://tcga-data.nci.nih.gov/). After sample barcode screening, 559 were miRNA-mRNA matched samples, of which 18 were distributed in the alveolar crest, 30 in the root of the tongue, 22 in the buccal mucosa, 67 in the mouth floor, 8 in the hard palate, 9 in the laryngeal pharynx, 138 in the larynx, 3 in the lip, 38 in the oral cavity, 156 in the tongue, 10 in the oropharynx, 45 in the tonsil and 15 from an unclear location. Only the 138 samples from the larynx were used in our study.

## Data preprocessing

For the data from Affy platform, the raw data in CEL. files were preprocessed using the oligo package (version 1.41.1; *Carvalho & Irizarry, 2010*) in R (version 3.4.1; *R Development Core Team, 2017*), including data transformation, missing value imputation with median, background correction with MAS method and quantile normalization.

For the data from Agilent and Illumina platforms, the raw data in TXT. files were preprocessed using the Linear Models for Microarray Data (LIMMA) package (version 3.34.0; *Ritchie et al., 2015*) in R, including data log2 transformation and median normalization.

The data (FPKM, fragment per kilobase per million mapped reads) from TCGA were quantile normalized using the preprocessCore package (version 1.40.0; *Bolstad, 2019*) in R.

## Differential expression analysis

The differentially expressed lncRNAs (DELs) and miRNAs (DEMs) between LSCC and normal controls were identified using the LIMMA method in R from their two included microarray datasets (lncRNA: GSE59652 and GSE84957; miRNA: gSE70289 and GSE62819). The *p*-value <0.05 and |logFC(fold change) | > 0.263 were set as the cut-off points. The overlap in the above two datasets was used for the following analysis of lncRNAs and miRNAs, respectively.

The differentially expressed genes (DEGs) between LSCC and normal controls were identified using the MetaDE.ES function in MetaDE package (version 1.0.5, https://cran.r-project.org/web/packages/MetaDE/) of R from its four included microarray datasets (GSE51985, GSE84957, GSE59102 and GSE58911). The *p*-value <0.05 and false discovery rate (FDR) <0.05 were set as the cut-off points. The DEGs with the same expression trend ($tau^2$ statistic = 0, *p*-value of Chi-square based Q-test >0.05) in the four datasets were selected for the following analysis.

Wilcoxon signed-rank test (http://www.bioconductor.org/help/search/index.html?q=wilcox.test/) was used to screen the DMRs between LSCC and normal controls. $P < 0.05$ was set as the threshold value. Human annotation data were retrieved from GENCODE

Release 19 (GRCh37.p13) (https://www.gencodegenes.org/human/release_19.html). The sequences of miRNAs, lncRNAs and mRNAs in the corresponding platform GPL8490 were blasted with the GRCh37.p13 to obtain the differentially methylated miRNAs, lncRNAs and mRNAs, which were then overlapped with the DELs, DEMs and DEGs to screen methylated-related DELs, DEMs and DEGs, respectively.

## CeRNA regulatory network construction

Three reliable online databases, including miRcode (version 11; http://www.mircode.org/), starBase (version 2.0; http://starbase.sysu.edu.cn/index.php) (*Li et al., 2014*) and DIANA-LncBase (version 2.0; http://carolina.imis.athena-innovation.gr/diana_tools/web/index.php?r=lncbasev2/index-predicted) (*Paraskevopoulou et al., 2013*) were used to screen the interactions between lncRNAs and miRNAs. The union of these three datasets was used for the following analysis. The target genes of miRNAs that were linked to the lncRNAs were predicted using four frequently used algorithms, including TargetScan (version 7.2; http://www.targetscan.org/vert_71/) (*Agarwal et al., 2015*), miRBase (version 22; https://www.ebi.ac.uk/enright-srv/microcosm/htdocs/targets/v5/) (*Griffithsjones et al., 2005*), miRanda (version 1.9; http://www.microrna.org/microrna/home.do/) (*John et al., 2005*) and miRTarBase (version 7.0; http://mirtarbase.mbc.nctu.edu.tw/php/index.php) (*Chou et al., 2017*). The target genes predicted by at least two databases and a negative association with miRNAs were retained. The lncRNA-miRNA and miRNA-mRNA interactions were integrated to construct the ceRNA network, which was visualized using Cytoscape software (version 3.4; *Shannon et al., 2001-2008*) (*Kohl, Wiese & Warscheid, 2011*).

## Function enrichment analysis

The Database for Annotation, Visualization and Integrated Discovery (DAVID) online tool (version 6.8; http://david.abcc.ncifcrf.gov) (*Da, Sherman & Lempicki, 2009*) was used for Gene Ontology (GO) terms [including molecular function (MF), biological process (BP) and cellular component (CC) categories] and Kyoto encyclopedia of genes and genomes (KEGG) pathway enrichment analyses of genes in the ceRNA network. *P*-value <0.05 was set as the cut-off value.

## Clinical associations of lncRNAs, miRNAs and mRNAs in the ceRNA network

The expression levels of lncRNAs, miRNAs and mRNAs in the ceRNA network were downloaded from the TCGA data. Univariate Cox regression analysis was performed to screen for the prognosis-related (including overall survival, OS; and recurrence-free survival, RFS) lncRNAs, miRNAs and mRNAs using the survival package (version 2.40.1; https://cran.r-project.org/package=survival), which was used to construct the prognosis-related ceRNA network. The samples were divided into two groups based on the expression of each lncRNA, miRNA and mRNA: a low expression group (<median) and a high expression (>median) group. The Kaplan–Meier method with the log-rank test was used to estimate the difference in OS and RFS between the high and low expression groups. $P < 0.05$ was considered statistically significant. Furthermore, multivariate

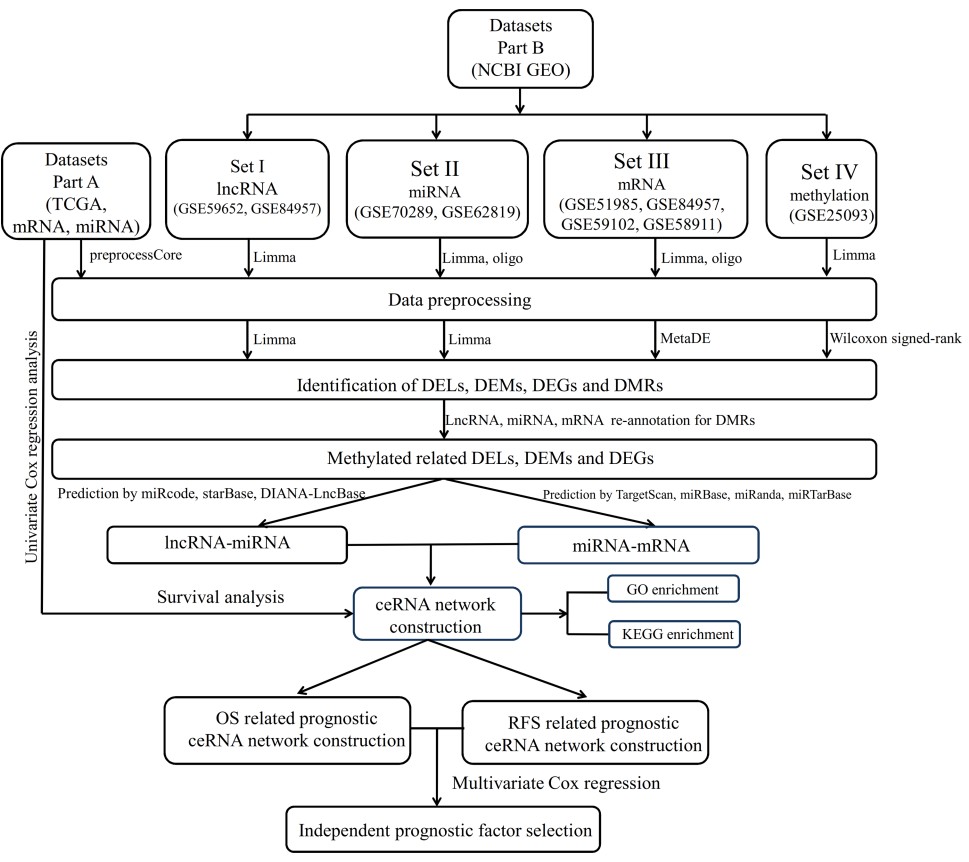

**Figure 1  The data analysis workflow.**

Cox regression analysis was also performed using the survival package (version 2.40.1; *Therneau & Lumley, 2019*) to evaluate the prognostic independence of lncRNAs, miRNAs and mRNAs. The association of nodes in the prognosis-related ceRNA network with other clinical characteristics was also analyzed using the multiple linear regression model (https://stat.ethz.ch/R-manual/R-patched/library/stats/html/lm.html) in R.

## RESULTS

### Differential expression analysis

The data analysis workflow is displayed in Fig. 1. After data normalization (Supplemental Information 1–8), the DELs, DEMs and DEGs between LSCC and normal samples were screened according the stated thresholds. The results showed 306 (156 downregulated and 150 upregulated) and 396 (252 downregulated and 144 upregulated) DELs were identified in the datasets of GSE59652 (Fig. 2A) and GSE84957 (Fig. 2B) (Supplemental Information 9), respectively. After comparison, 40 DELs were found to be shared in these two datasets, including six upregulated and 20 downregulated with the consistent expression trend (Fig. 3A) (Supplemental Information 9); a total of 1,307 (765 downregulated and 542

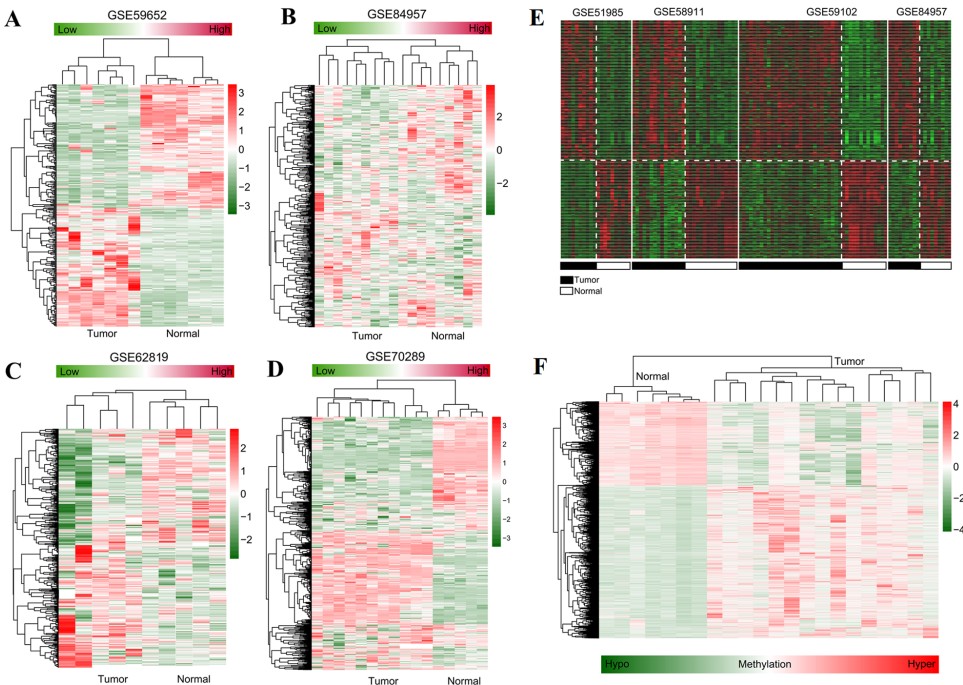

**Figure 2** **Hierarchical clustering and heat map analysis.** (A–B) heat map for differentially expressed lncRNAs identified in GSE59652 (A) and GSE84957 (B) datasets; (C–D), heat map for differentially expressed miRNAs identified in GSE62819 (C) and GSE70289 (D) datasets; (E) heat map for differentially expressed genes identified by meta-analysis of GSE51985, GSE84957, GSE59102 and GSE58911 datasets; (F) heat map for differentially methylated regions identified in the GSE25093 dataset. The datasets of laryngeal squamous cell carcinoma collected from Gene Expression Omnibus database. Red, high expression (hyper-methylation); green, low expression (hypo-methylation).

upregulated) and 491 (126 downregulated and 365 upregulated) DEMs were identified in the datasets GSE62819 (Fig. 2C) and GSE70289 (Fig. 2D), respectively (Supplemental Information 9). After comparison, 443 DEMs were found to be common in these two datasets, among which 152 upregulated and 63 downregulated DEMs were shown to have a consistent expression trend (Fig. 3B) (Supplemental Information 9); 2,975 DEGs were found to display the similar expression trend in four mRNA expression profiles GSE51985, GSE84957, GSE59102 and GSE58911 (Fig. 2E) (Supplemental Information 9); and 4,567 DMRs were identified in the LSCC genome of GSE25093 dataset, including 1616 hypomethylated and 2,951 hypermethylated (Fig. 2F) (Supplemental Information 9). After GENCODE annotation and blast analysis, 122 lncRNAs, but no miRNAs were found to be located in DMRs. Subsequently, the lncRNAs and mRNAs in DMRs were overlapped with their expression level data above to obtain the methylation-related DELs and DEGs. Consequently, five DELs (TMEM51-AS1, HCP5, SND1-IT1, RUSC1-AS1 and LINC00324) were screened (Fig. 3C). Among these DELs, only the expression and methylation levels of lncRNA TMEM51-AS1 (Figs. 3D–3F) were opposite, indicating its expression may be regulated by methylation. These methylation-related genes were used to construct the ceRNA network.

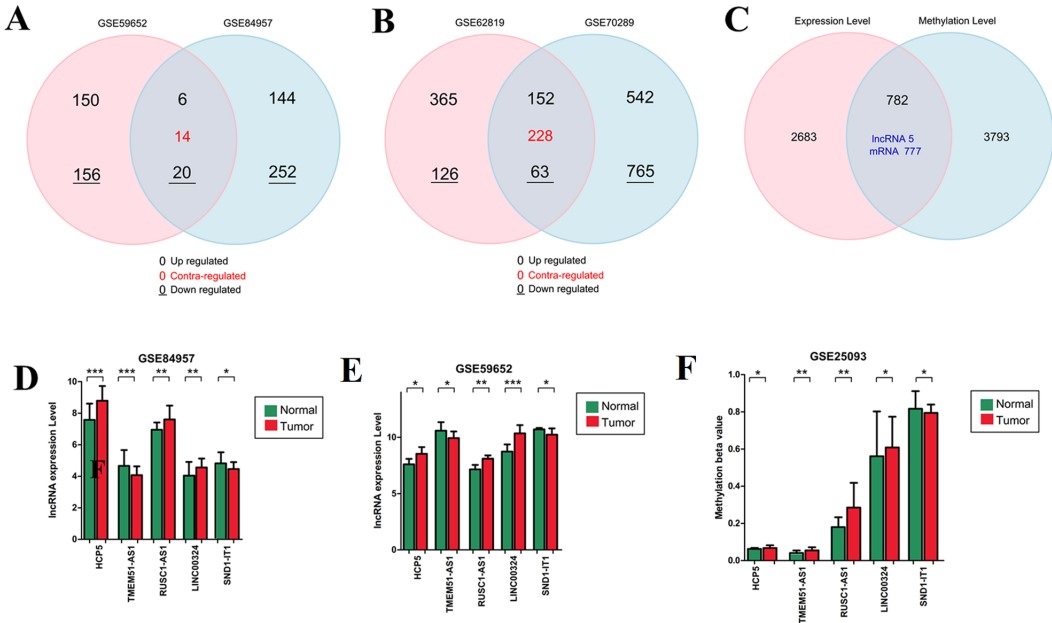

**Figure 3  Overlapped genes identification.** Venn diagram drawing to display the overlap of differentially expressed lncRNAs (A) and miRNAs (B) in different datasets of laryngeal squamous cell carcinoma collected from Gene Expression Omnibus database and their overlap with differentially methylated regions (C) to screen methylation related lncRNAs and miRNAs. The expression (D–E) and methylated (F) levels of overlapped lncRNAs are displayed in a histogram. *$p < 0.05$; **$p < 0.01$; ***$p < 0.001$. Contra-regulated: the expression trend of lncRNAs or miRNAs was different in two datasets. Upregulated or downregulated: lncRNAs or miRNAs exhibited the similar expression trend in two datasets, high or down expressed.

## CeRNA network construction

Twenty-four interaction pairs between five DELs and 14 DEMs were predicted using miRcode, starBase and DIANA-LncBase databases (Table 1). The expression trends of these DELs and DEMs were opposite. Subsequently, the target genes of these 14 DEMs were predicted using four algorithms, with the resultant interaction pairs of 700 in TargetScan, 486 in miRBase, 341 in miRanda and 268 in miRTarBase. A total of 404 interaction pairs were ultimately left due to prediction by at least two databases and a negative association between them. These interaction pairs between DELs and DEMs, and between DEMs and DEGs were used to construct a ceRNA network, which contained 258 nodes (five DELs, 11 DEMs and 242 DEGs) (Fig. 4). In this network, TMEM51-AS1 functioned as a ceRNA to regulate SNX21 (sorting nexin family member 21) and TRAPPC10 (trafficking protein particle complex 10) by sponging miR-106b; LINC00324 and RUSC1-AS1 acted as ceRNAs to regulate SPRY4 (sprouty RTK signaling antagonist 4), PAWR (pro-apoptotic WT1 regulator), MICAL2 (microtubule associated monooxygenase, calponin and LIM domain containing 2) and SLC39A14 (solute carrier family 39 member 14) by sponging miR-16; RUSC1-AS1 regulated SCG5 (SCG5 secretogranin V) and PRDM5 (PR/SET domain 5) by competitively binding to miR-10; RUSC1-AS1 also served as ceRNAs for ZFP1 (ZFP1 zinc finger protein) by binding to miR-7; HCP5 could interact with miR-143 to regulate RRM2 (ribonucleotide reductase regulatory subunit M2).
**Table 1  Interaction relationship between lncRNA and miRNAs.**

| lncRNA | miRNA |
|---|---|
| HCP5 | hsa-miR-10, hsa-miR-16, hsa-miR-186, hsa-miR-214, hsa-miR-7, hsa-miR-641, hsa-miR-143, hsa-miR-4770, hsa-miR-216b, hsa-miR-876 |
| LINC00324 | hsa-miR-143, hsa-miR-16, hsa-miR-214, hsa-miR-216b, hsa-miR-4770 |
| RUSC1-AS1 | hsa-miR-214, hsa-miR-10, hsa-miR-16, has-miR-216b, hsa-miR-7 |
| TMEM51-AS1 | hsa-miR-106b, hsa-miR-765 |
| SND1-IT1 | hsa-miR-708, hsa-miR-4306 |

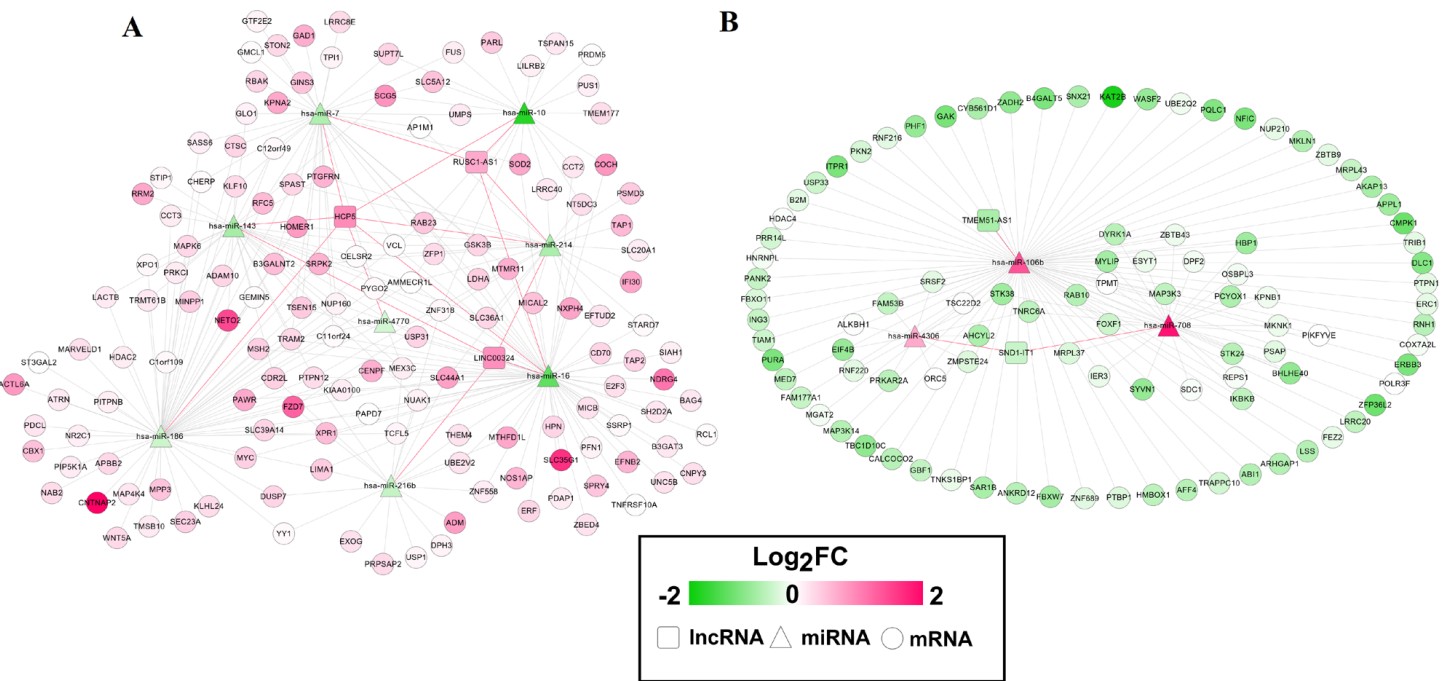

**Figure 4  Competing endogenous RNAs (ceRNAs) interaction network of lncRNA-miRNA-mRNA in laryngeal squamous cell carcinoma.** (A) interaction pairs among upregulated lncRNAs, downregulated miRNAs and upregulated mRNAs; (B) interaction pairs among downregulated lncR-NAs, upregulated miRNAs and downregulated mRNAs. Square nodes represent lncRNAs; triangle nodes represent miRNAs; round nodes represent mRNAs. Edges represent the possible associations between lncRNAs, miRNAs and mRNAs. Red, upregulated; green, downregulated. Red line, the interaction between lncRNAs and miRNAs; greyish line, the interaction between miRNA and mRNAs.

## Function enrichment analysis

The DEGs in the ceRNA network was subjected to DAVID to predict their potential functions in LSCC. The results showed that 17 significant GO BP terms were enriched, including GO:0042981~ regulation of apoptosis (PAWR), GO:0015031~ protein transport (SNX21; SCG5), cell cycle (PRDM5) and GO:0043407~ negative regulation of MAP kinase activity 4 (SPRY4). Six KEGG pathways were also enriched, including hsa05210: colorectal cancer, hsa04210:Apoptosis and hsa05205:Proteoglycans in cancer (Table 2).

**Table 2  Function enrichment analysis for the genes in ceRNA network.**

| Category | Term | *P*-value | Genes |
|---|---|---|---|
| Biology process | GO:0006793∼ phosphorus metabolic process | 0.00122 | STK38, SLC20A1, ERBB3, NUAK1, MKNK1, ABI1, PIP5K1A, TRIB1, MAP3K3, SRPK2, MINPP1, ADAM10, STK24, MSH2, PRKCI, PKN2, PTPN12, GAK, MAP4K4, MTMR11, MAPK6, GSK3B, DYRK1A, PTPN1, MAP3K14, ERC1, IKBKB, DUSP7 |
| | GO:0006796∼ phosphate metabolic process | 0.00122 | STK38, SLC20A1, ERBB3, NUAK1, MKNK1, ABI1, PIP5K1A, TRIB1, MAP3K3, SRPK2, MINPP1, ADAM10, STK24, MSH2, PRKCI, PKN2, PTPN12, GAK, MAP4K4, MTMR11, MAPK6, GSK3B, DYRK1A, PTPN1, MAP3K14, ERC1, IKBKB, DUSP7 |
| | GO:0006468∼ protein amino acid phosphorylation | 0.00498 | SRPK2, ADAM10, STK38, STK24, NUAK1, ERBB3, PRKCI, PKN2, MKNK1, ABI1, TRIB1, GAK, MAP4K4, MAPK6, MAP3K3, GSK3B, DYRK1A, IKBKB, ERC1, MAP3K14 |
| | GO:0016310∼ phosphorylation | 0.00801 | SRPK2, ADAM10, STK38, ERBB3, MSH2, STK24, NUAK1, PRKCI, PKN2, MKNK1, ABI1, PIP5K1A, TRIB1, GAK, MAP4K4, MAPK6, MAP3K3, GSK3B, DYRK1A, IKBKB, ERC1, MAP3K14 |
| | GO:0008104∼ protein localization | 0.01176 | STON2, SEC23A, XPO1, AP1M1, NUP160, PRKCI, CENPF, TMSB10, TRAM2, TAP2, GSK3B, NUP210, TAP1, PIKFYVE, SNX21, RAB23, SCG5, SUPT7L, SAR1B, RAB10, ERC1, KPNA2, KPNB1 |
| | GO:0043407∼ negative regulation of MAP kinase activity | 0.00991 | STK38, PTPN1, SPRY4, DUSP7 |
| | GO:0042981∼ regulation of apoptosis | 0.0165 | DLC1, DPF2, IER3, ING3, SYVN1, ERBB3, MSH2, KLF10, PRKCI, AKAP13, CD70, PAWR, SOD2, TNFRSF10A, BAG4, TIAM1, GSK3B, GLO1, APBB2, IKBKB, MYC |
| | GO:0043067∼ regulation of programmed cell death | 0.0181 | DLC1, DPF2, IER3, ING3, SYVN1, ERBB3, MSH2, KLF10, PRKCI, AKAP13, CD70, PAWR, SOD2, TNFRSF10A, BAG4, TIAM1, GSK3B, GLO1, APBB2, IKBKB, MYC |
| | GO:0015031∼ protein transport | 0.0188 | STON2, SEC23A, XPO1, AP1M1, NUP160, PRKCI, CENPF, TRAM2, TAP2, GSK3B, NUP210, TAP1, SNX21, RAB23, SCG5, SAR1B, RAB10, ERC1, KPNA2, KPNB1 |
| | GO:0010941∼ regulation of cell death | 0.0188 | DLC1, DPF2, IER3, ING3, SYVN1, ERBB3, MSH2, KLF10, PRKCI, AKAP13, CD70, PAWR, SOD2, TNFRSF10A, BAG4, TIAM1, GSK3B, GLO1, APBB2, IKBKB, MYC |
| | GO:0045184∼ establishment of protein localization | 0.0204 | STON2, SEC23A, XPO1, AP1M1, NUP160, PRKCI, CENPF, TRAM2, TAP2, GSK3B, NUP210, TAP1, SNX21, RAB23, SCG5, SAR1B, RAB10, ERC1, KPNA2, KPNB1 |
| | GO:0008219∼ cell death | 0.0211 | FUS, DPF2, DLC1, IER3, MICB, ERBB3, MSH2, AKAP13, RNF216, PAWR, ITPR1, SOD2, TNFRSF10A, BAG4, UNC5B, TIAM1, SIAH1, MYC, SPAST |
| | GO:0010033∼ response to organic substance | 0.0217 | ADAM10, KAT2B, ERBB3, MSH2, KLF10, PRKCI, CALCOCO2, APPL1, TRIB1, B2M, HDAC4, PRKAR2A, SDC1, HDAC2, ADM, TAP2, CTSC, PTPN1, MYC |
| | GO:0016265∼ death | 0.0225 | FUS, DPF2, DLC1, IER3, MICB, ERBB3, MSH2, AKAP13, RNF216, PAWR, ITPR1, SOD2, TNFRSF10A, BAG4, UNC5B, TIAM1, SIAH1, MYC, SPAST |
| | GO:0044265∼ cellular macromolecule catabolic process | 0.0228 | ADAM10, SYVN1, USP1, RNH1, UBE2V2, RNF216, MYLIP, UBE2Q2, ZFP36L2, FBXW7, GMCL1, PSMD3, ZMPSTE24, SIAH1, PCYOX1, USP33, MYC, FBXO11, USP31 |
**Table 2** (*continued*)

| Category | Term | *P*-value | Genes |
|---|---|---|---|
| | GO:0007049~ cell cycle | 0.0405 | E2F3, KAT2B, MSH2, PAPD7, CENPF, APPL1, GAK, SASS6, MAPK6, GSK3B, PRDM5, PSMD3, ZNF318, HBP1, SIAH1, APBB2, KPNA2, MYC, SPAST |
| | GO:0009057~ macromolecule catabolic process | 0.0427 | ADAM10, SYVN1, USP1, RNH1, UBE2V2, RNF216, MYLIP, UBE2Q2, ZFP36L2, FBXW7, GMCL1, PSMD3, ZMPSTE24, SIAH1, PCYOX1, USP33, MYC, FBXO11, USP31 |
| KEGG pathway | hsa05210:Colorectal cancer | 0.0377 | MSH2, GSK3B, APPL1, MYC, FZD7 |
| | hsa04210:Apoptosis | 0.0421 | TNFRSF10A, PRKAR2A, EXOG, IKBKB, MAP3K14 |
| | hsa00562:Inositol phosphate metabolism | 0.0477 | MINPP1, TPI1, PIKFYVE, PIP5K1A |
| | hsa05169:Epstein-Barr virus infection | 0.00175 | POLR3F, XPO1, HDAC4, GTF2E2, HDAC2, GSK3B, PSMD3, MAP3K14, IKBKB, MYC |
| | hsa05166:HTLV-I infection | 0.0122 | WNT5A, XPO1, E2F3, KAT2B, MAP3K3, GSK3B, MAP3K14, IKBKB, MYC, FZD7 |
| | hsa05205:Proteoglycans in cancer | 0.0267 | WNT5A, EIF4B, SDC1, TIAM1, ERBB3, MYC, FZD7, ITPR1 |

## Clinical associations of lncRNAs, miRNAs and mRNAs in the ceRNA network

In the 138 miRNA-mRNA matched samples of TCGA data, 114 had OS and 82 had RFS information. Univariate Cox regression analysis in these 138 samples showed that 32 RNAs were significantly associated with OS, including 1 DEL (RUSC1-AS1), 2 DEMs (hsa-miR-16 and hsa-miR-10) and 29 DEGs (i.e., PAWR, SCG5, SPRY4, MICAL2, SNX21, TRAPPC10 and SLC39A14); while 25 RNAs were associated with RFS, including one DEL (LINC00324), three DEMs (hsa-miR-16, hsa-miR-10 and hsa-miR-7) and 21 DEGs (i.e., PRDM5, SCG5, SPRY4, MICAL2 and ZFP1) (Table 3). The OS and RFS related ceRNA networks were extracted independently as shown in Figs. 5A and 5B.

Subsequently, multivariate Cox regression showed TRAPPC10 and SLC39A14 were independent factors for OS; RRM2 was an independent factor for RFS (Table 4). Although SOD2, SLC44A1 and THEM4 were also screened to be significant, their hazard ratios (HR) were not consistent with the expected according to their expression levels. Combined with the univariate results, we suggested TRAPPC10 and SLC39A14 related ceRNA axes (TMEM51-AS1-miR-106-TRAPPC10; RUSC1-AS1-miR-16-SLC39A14) may be especially important. The Kaplan–Meier curve of these lncRNAs, miRNAs and mRNAs were drawn. As expected, the low expression of miR-16 (Fig. 5D) was associated with poor prognosis and the high expression of RUSC1-AS1 (Fig. 5C), SLC39A14 (Fig. 5E) and TRAPPC10 (Fig. 6A) was associated with shorter OS.

Furthermore, OS- and RFS-related DELs, DEMs and DEGs were also analyzed to investigate their associations with other clinical characteristics of LSCC to further confirm their importance. The results showed that RUSC1-AS1 was significantly associated with Pathologic N; OS- and RFS-related SPRY4 was associated with Pathologic M; OS-related MICAL2 was associated with Pathologic N and Pathologic stage; RFS-related ZFP1 and SLC39A14 were associated with Pathologic N; OS-related SNX21 and RFS-related SCG5

**Table 3   Prognosis related lncRNAs, miRNAs and mRNAs in ceRNA network.**

| | Overall survival | | | | Recurrence free survival | | |
|---|---|---|---|---|---|---|---|
| | RNA | exp(coef) | *p* | | RNA | exp(coef) | *p* |
| miRNA | hsa-miR-16 | 0.506 | 0.00048 | miRNA | hsa-miR-10 | 0.678 | 0.0135 |
| | hsa-miR-10 | 0.653 | 0.016 | | hsa-miR-16 | 0.523 | 0.00355 |
| lncRNA | RUSC1-AS1 | 1.09 | 0.01 | | hsa-miR-7 | 1.75 | 0.011 |
| mRNA | ADAM10 | 2.01 | 0.0435 | lncRNA | LINC00324 | 1.38 | 0.0205 |
| | AHCYL2 | 0.739 | 0.0315 | mRNA | AFF4 | 2.43 | 0.036 |
| | CNPY3 | 0.602 | 0.048 | | CELSR2 | 0.576 | 0.041 |
| | DLC1 | 1.56 | 0.025 | | ERBB3 | 0.392 | 0.041 |
| | E2F3 | 0.661 | 0.0355 | | LRRC8E | 0.577 | 0.029 |
| | FUS | 0.531 | 0.0345 | | MAP4K4 | 2.12 | 0.0475 |
| | HPN | 0.878 | 0.036 | | MICAL2 | 1.76 | 0.0115 |
| | ITPR1 | 2.19 | 0.036 | | NXPH4 | 0.702 | 0.0265 |
| | LRRC40 | 2.56 | 0.042 | | PCYOX1 | 0.484 | 0.0445 |
| | MICAL2 | 1.49 | 0.0325 | | PRDM5 | 1.64 | 0.0135 |
| | NXPH4 | 0.722 | 0.032 | | PTBP1 | 8.8 | 0.047 |
| | PAWR | 1.57 | 0.037 | | PTPN1 | 3.26 | 0.038 |
| | PRPSAP2 | 0.563 | 0.045 | | PYGO2 | 0.252 | 0.036 |
| | PTPN12 | 2.19 | 0.0485 | | RRM2 | 2.32 | 0.043 |
| | PUS1 | 0.668 | 0.0385 | | SCG5 | 1.84 | 0.00065 |
| | PYGO2 | 0.334 | 0.0485 | | SDC1 | 0.459 | 0.042 |
| | RAB10 | 1.59 | 0.0295 | | SLC39A14 | 1.85 | 0.042 |
| | SAR1B | 1.62 | 0.039 | | SLC44A1 | 0.445 | 0.011 |
| | SCG5 | 1.4 | 0.0295 | | SPRY4 | 2.28 | 0.007 |
| | SLC39A14 | 1.78 | 0.027 | | ST3GAL2 | 2.34 | 0.033 |
| | SNX21 | 0.502 | 0.048 | | THEM4 | 0.278 | 0.006 |
| | SOD2 | 0.736 | 0.038 | | ZFP1 | 3.35 | 0.032 |
| | SPRY4 | 1.98 | 0.023 | | | | |
| | ST3GAL2 | 1.87 | 0.0345 | | | | |
| | TAP2 | 0.68 | 0.0425 | | | | |
| | TCFL5 | 0.613 | 0.033 | | | | |
| | TRAPPC10 | 0.324 | 0.037 | | | | |
| | TSC22D2 | 1.36 | 0.0295 | | | | |
| | TSEN15 | 1.58 | 0.0415 | | | | |

were associated with gender (Table 5). These findings implied SPRY4, MICAL2, ZFP1, SNX21 and SCG5 related ceRNAs (LINC00324/RUSC1-AS1-miR-16-SPRY4/MICAL2, RUSC1-AS1-miR-7-ZFP1, TMEM51-AS1-miR-106-SNX21, RUSC1-AS1-miR-10-SCG5) were also crucial for LSCC. The Kaplan–Meier curve of SNX21 is shown in Fig. 6B and the other DEGs are displayed in Figs. S1 and S2.

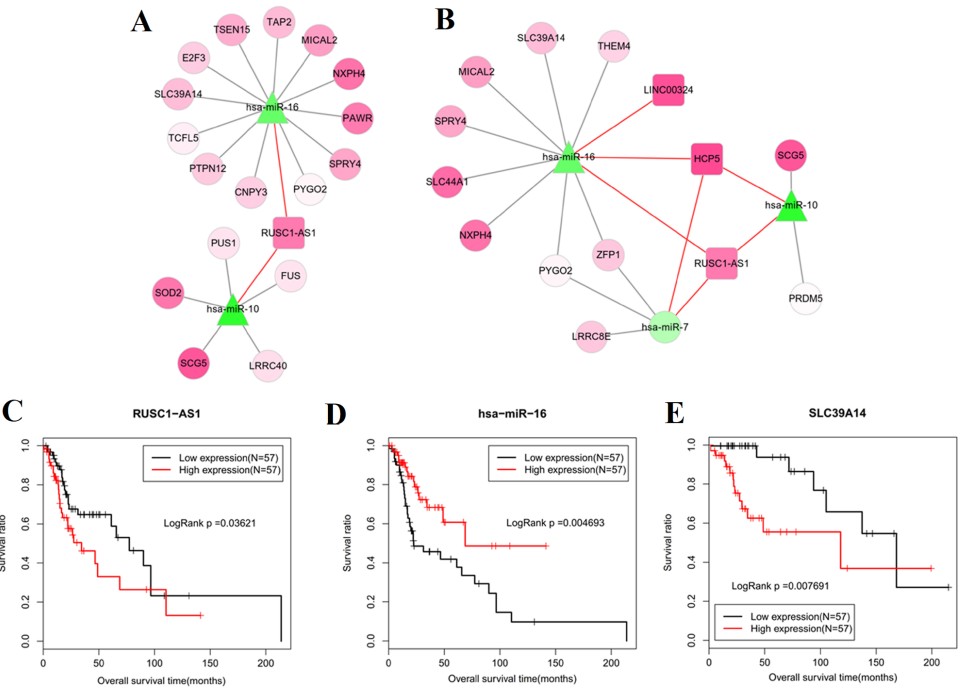

**Figure 5** **Prognosis related competing endogenous RNAs (ceRNAs) interaction axes.** (A) lncRNA-miRNA-mRNA network for overall survival; (B) lncRNA-miRNA-mRNA network for recurrence free survival. Square nodes represent lncRNAs; triangle nodes represent miRNAs; round nodes represent mRNAs. Edges represent the possible associations between lncRNAs, miRNAs and mRNAs. Red, upregulated; green, downregulated.; Kaplan–Meier analysis of the lncRNA (C), miRNA (D) and mRNA (E) of crucial ceRNA axis in which all lncRNA, miRNA and mRNA were prognosis-related and mRNA was an independent prognostic factor.

## DISCUSSION

Although epigenetics modification has been shown to trigger silencing or overexpression of lncRNAs in cancer (*Dong et al., 2017*; *Guo et al., 2018b*; *Zhou et al., 2018*), the aberrant methylation-mediated expression changes of lncRNAs remain unclear in LSCC. We, for the first time, found that the downregulation of lncRNA TMEM51-AS1 may be mediated by hyper-methylation. Few studies investigated the roles of TMEM51-AS1 in cancer except one study indicated downregulated TMEM51-AS1 was significantly correlated with poor OS in chromophobe renal cell carcinoma (*He et al., 2016*). In the present study, we predicted that TMEM51-AS1 might function as a ceRNA to regulate SNX21 and TRAPPC10 through sponging miR-106b. Evidence demonstrated that miR-106b was up-regulated in LSCC (*Lu et al., 2014*; *Xing et al., 2014*), which was also confirmed in our microarray study. miR-106b was reported to promote the proliferation and invasion of LSCC cells by targeting RUNX3 (*Ying et al., 2013*), while induce cell cycle G0/G1 arrest by inhibiting tumor suppressor RB (*Cai, Wang & Bao, 2011*). Although no study revealed the roles of SNX21 in cancer, its family genes, such as SNX1 (*Zhan et al., 2018*), SNX5 (*Jitsukawa et al., 2017*) and SNX9 (*Bendris et al., 2016*) were suggested to be tumor suppressor related. Therefore, SNX21 may be theoretically downregulated in LSCC by miR-106b. Consistent with this hypothesis,

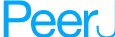

**Table 4  Independent prognostic factors for LSCC by multivariate Cox regression.**

| | | OS | | | | | RFS | | | |
|---|---|---|---|---|---|---|---|---|---|---|
| ID | *P*-value | HR | 95% CI | | ID | *P*-value | HR | 95% CI | | |
| | | | Lower limit | Upper limit | | | | Lower limit | Upper limit | |
| TRAPPC10 | **0.0106** | 0.0941 | 0.01535 | 0.5768 | SLC44A1 | **0.0055** | 0.1719 | 0.0496 | 0.5958 | |
| Alcohol | **0.0252** | 0.0391 | 0.00229 | 0.668 | THEM4 | **0.0101** | 0.2215 | 0.07023 | 0.6988 | |
| SLC39A14 | **0.0289** | 9.37 | 1.26 | 69.8 | RRM2 | **0.0151** | 34.8562 | 1.99009 | 610.503 | |
| SOD2 | **0.0456** | 2.748 | 1.01992 | 7.4038 | Age | 0.0875 | 0.06502 | 0.00283 | 1.494 | |
| SCG5 | 0.0515 | 8.6 | 0.986 | 75.1 | T | 0.1882 | 0.09721 | 0.00302 | 3.129 | |
| Grade | 0.0536 | 0.0053 | 0.000025 | 1.09 | Stage | 0.1883 | 14.2194 | 0.27247 | 742.056 | |
| MICAL2 | 0.0872 | 0.141 | 0.0149 | 1.33 | ZFP1 | 0.2108 | 3.5709 | 0.48649 | 26.2105 | |
| RUSC1-AS1 | 0.0894 | 1.1434 | 0.97957 | 1.3347 | LINC00324 | 0.2134 | 1.5429 | 0.77921 | 3.0549 | |
| Gender | 0.1057 | 0.002 | 1.08E−06 | 3.72 | PCYOX1 | 0.2206 | 0.16586 | 0.009362 | 2.939 | |
| CNPY3 | 0.1069 | 43.1 | 0.444 | 4170 | CELSR2 | 0.2233 | 0.12475 | 0.004376 | 3.556 | |
| PUS1 | 0.1534 | 0.0927 | 0.00354 | 2.43 | NXPH4 | 0.2429 | 0.58688 | 0.239935 | 1.435 | |
| HPN | 0.1538 | 0.398 | 0.112 | 1.41 | Gender | 0.2507 | 0.11327 | 0.002754 | 4.658 | |
| hsa-mir-16-2 | 0.1626 | 0.5574 | 0.24539 | 1.266 | PTPN1 | 0.2567 | 17.1232 | 0.126445 | 2318.83 | |
| Age | 0.1641 | 0.0323 | 0.000256 | 4.07 | AFF4 | 0.3093 | 17.52882 | 0.070156 | 4379.665 | |
| PTPN12 | 0.204 | 42.7 | 0.13 | 14000 | PYGO2 | 0.3218 | 0.06118 | 0.000243 | 15.379 | |
| ADAM10 | 0.245 | 0.136 | 0.00472 | 3.93 | SLC39A14 | 0.3303 | 1.5919 | 0.62436 | 4.0587 | |
| LRRC40 | 0.2467 | 0.0333 | 0.000106 | 10.5 | SPRY4 | 0.3574 | 1.6716 | 0.55973 | 4.9922 | |
| Stage | 0.2589 | 185 | 0.0215 | 158000 | Grade | 0.4953 | 0.24206 | 0.004104 | 14.277 | |
| RAB10 | 0.2779 | 0.0066 | 7.71E-07 | 57.1 | MAP4K4 | 0.4973 | 0.21996 | 0.002775 | 17.436 | |
| T | 0.2814 | 0.0214 | 0.000020 | 23.3 | hsa-mir-16-2 | 0.50105 | 0.7001 | 0.24778 | 1.978 | |
| TSC22D2 | 0.3088 | 1.7322 | 0.60129 | 4.99 | ERBB3 | 0.6193 | 2.56113 | 0.06268 | 104.645 | |
| FUS | 0.3262 | 0.0128 | 2.11E-06 | 77.1 | PTBP1 | 0.629 | 8.46884 | 0.00146 | 49149.03 | |
| DLC1 | 0.3279 | 0.168 | 0.00472 | 5.99 | tobacco | 0.6901 | 0.59073 | 0.04442 | 7.856 | |
| PRPSAP2 | 0.328 | 0.0598 | 0.000212 | 16.9 | SCG5 | 0.7040 | 1.2036 | 0.46263 | 3.1314 | |
| TSEN15 | 0.3295 | 2.1853 | 0.45397 | 10.5197 | PRDM5 | 0.7108 | 1.45392 | 0.20106 | 10.514 | |
| ST3GAL2 | 0.3643 | 1.4916 | 0.62882 | 3.5381 | N | 0.7903 | 0.69963 | 0.05030 | 9.731 | |
| PYGO2 | 0.4608 | 6.86 | 0.0412 | 1140 | LRRC8E | 0.7914 | 0.70897 | 0.05545 | 9.065 | |
| E2F3 | 0.4736 | 2.69 | 0.18 | 40.2 | MICAL2 | 0.8244 | 1.2676 | 0.15616 | 10.289 | |
| N | 0.5037 | 0.203 | 0.00188 | 21.8 | SDC1 | 0.8476 | 1.1355 | 0.31079 | 4.1489 | |
| SPRY4 | 0.5551 | 1.3026 | 0.54132 | 3.1346 | hsa-mir-7-2 | 0.8556 | 1.1305 | 0.30175 | 4.2351 | |
| TAP2 | 0.574 | 0.7112 | 0.21679 | 2.3332 | hsa-mir-10a | 0.9186 | 0.9692 | 0.53189 | 1.7661 | |
| NXPH4 | 0.6486 | 0.784 | 0.276 | 2.23 | ST3GAL2 | 0.9385 | 0.9482 | 0.24524 | 3.666 | |
| TCFL5 | 0.6954 | 0.7436 | 0.16866 | 3.278 | Alcohol | 0.9937 | 1.01803 | 0.01187 | 87.316 | |
| Tobacco | 0.717 | 1.67 | 0.105 | 26.5 | | | | | | |
| SNX21 | 0.8253 | 0.8703 | 0.25359 | 2.9871 | | | | | | |
| hsa-mir-10a | 0.8392 | 0.9451 | 0.54739 | 1.6316 | | | | | | |
| PAWR | 0.842 | 1.7 | 0.00918 | 315 | | | | | | |
| SAR1B | 0.9106 | 1.25 | 0.0244 | 64.3 | | | | | | |
| ITPR1 | 0.9121 | 1.19 | 0.0539 | 26.3 | | | | | | |
| AHCYL2 | 0.9767 | 1.06 | 0.0236 | 47.5 | | | | | | |

**Notes.**
*P*-value < 0.05 shown in bold.

 

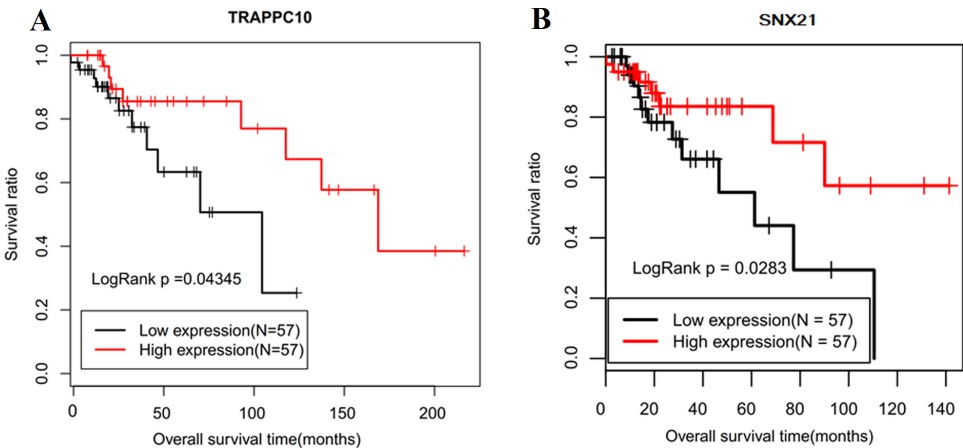

**Figure 6    Kaplan–Meier curve of lncRNA TMEM51-AS1 ceRNA related mRNAs.** (A) TRAPPC10, which was an independent prognostic factor; (B) SNX21, which was overall survival related in univariate Cox regression analysis.

**Table 5    Clinical characteristics related to lncRNAs, miRNAs and mRNAs in prognostic ceRNA network.**

| Clinical characteristics | Significant related | | |
|---|---|---|---|
| | lncRNA | miRNA | mRNA |
| Age( ≥60/<60 y) | – | – | ADAM10, FUS, MICAL2,  LRRC8E |
| Gender(Male/Female) | – | – | DLC1, HPN, PTPN12, SNX21, ST3GAL2, LRRC8E, NXPH4, PTPN1, SCG5, |
| Alcohol use(Yes/No) | – | – | CNPY3, PYGO2, SLC39A14, TAP2 |
| Pathologic_M(M0/-) | – | – | ADAM10, CNPY3, E2F3, **SPRY4**, LRRC8E, MAP4K4, PTBP1, RRM2 |
| Pathologic_N(N0/N1/N2/N3/-) | RUSC1-AS1 | – | DLC1, FUS, MICAL2, SOD2, PCYOX1, RRM2, SLC39A14, ZFP1 |
| Pathologic_T(T1/T2/T3/T4/-) | – | – | LRRC40, PRPSAP2, SOD2, TSEN15,  CELSR2, PTPN1 |
| Pathologic_stage(I/II/III/IV/-) | – | – | MICAL2, PRPSAP2, SPRY4, CELSR2, MAP4K4, PCYOX1, PTPN1 |
| Grade(G1/G2/G3/G4) | – | - | AHCYL2, DLC1 , PCYOX1, PTPN1, RRM2 |
| Tobacco use(Reform/Current/Never) | – | – | HPN, RAB10, SAR1B, SOD2, TAP2 |

**Notes.**

Underlined genes were recurrence free survival related; the other genes were overall survival related. Bolded genes was both recurrence free and overall survival related.

our study showed that SNX21 was less expressed in LSCC tissues and patients with high expression of SNX21 had a higher OS rate. There was only one study to suggest the roles of TRAPPC10 until now and showed TRAPPC10 was an oncogenic driver to predict the poor prognosis for breast cancer patients (*Pongor et al., 2015*), which seemed to be contrast with our results, implying TRAPPC10 may be a new tumor suppressor gene for LSCC. The tumor inhibition effects of TRAPPC10 may be related to its potential to activate GTPase RAB11 (*Milev et al., 2018*) and the Rab coupling protein, the targeted deletion of which led to accelerated tumor onset (*Boulay et al., 2016*).

Furthermore, we identified several other ceRNA axes, although they were not methylation-related, including LINC00324/RUSC1-AS1-miR-16-SPRY4/MICAL2/SLC39A14, RUSC1-AS1-miR-10-SCG5 and RUSC1-AS1-miR-7-ZFP1. All these lncRNAs, miRNAs and mRNAs were significantly associated with OS and/or RFS, indicating these ceRNA axes may also be underlying therapeutic targets.

Although related report was rare, RUSC1-AS1 (*Jian et al., 2015*) and LINC00324 (*Militello et al., 2017*) had been indicated to be highly expressed in cancer cells, which were similarly confirmed in LSCC samples. Accumulating evidence also has proved the roles of miR-16, miR-7 and miR-10 in various types of cancer. miR-16 could be downregulated in tissue samples and cell lines of lung cancer (*Ke et al., 2013*) and osteosarcoma (*Jiao, Wang & Wang, 2018*). Ectopic expression of miR-16 inhibited cell proliferation, colony formation *in vivo* and, migration and invasion *in vitro* by regulating its target genes RAB23 and Smad3 (*Jiao, Wang & Wang, 2018*; *Zhang et al., 2018*). miR-10a was down-regulated in laryngeal epithelial premalignant lesions with increasing grade of dysplasia (*Hu et al., 2015*). Overexpression of miR-10a inhibited cell metastasis by regulating epithelial-to-mesenchymal transition (EMT) (*Liu et al., 2017b*). miR-7-5p was lower expressed in brain-metastatic lesions of breast cancer (*Hiroshi et al., 2013*) and the use of miR-7-5p mimics suppressed cell proliferation and induced apoptosis (*Shi et al., 2015*) via modulating the expression of Kruppel like factor 4. In agreement with these studies, we also found that these three miRNAs were less expressed (especially miR-10 and miR-7) in LSCC and negatively associated with OS and/or RFS. Although the downstream target genes of these miRNAs have been reported as above, their functions in LSCC remain poorly understood. We predicted that SPRY4/MICAL2/SLC39A14, SCG5 and ZFP1 may be the potential targets of miR-16, miR-10 and miR-7, respectively in LSCC, which had not been validated previously. Nevertheless, the studies on the molecular mechanisms of these DEGs may indirectly explain their potential interactions. The expression of SPRY4 was upregulated in testicular germ cell tumors (*Tian et al., 2018*). MICAL2 was a recently identified proto-oncogene, which increased cell proliferation to accelerate tumor growth, and promoted the expression of EMT-related proteins to increase cell metastasis (*Mariotti et al., 2016*; *Wang et al., 2018*). Immunohistochemical analysis showed the expression level of SLC39A14 was significantly higher in hepatocellular carcinoma tissues than that in adjacent tissues and negatively correlated with survival time (*Gartmann et al., 2018*). Also, the upregulation of SLC39A14 in tumor cells may be attributed to the loss of its interactive gene p53, a tumor suppressor (*Zhao et al., 2017*). Although there were no studies to discuss the roles of SCG5 in cancer, its family members secretogranin II and III have been seen to be overexpressed in prostate cancer (*Courel et al., 2014*) and small cell lung carcinoma (*Togayachi et al., 2017*), suggesting SCG5 may also be oncogenic for LSCC. Zinc finger proteins had also been observed to promote cell growth and metastasis in nasopharyngeal carcinoma (*Li et al., 2015*). In line with these findings, SPRY4, MICAL2, SLC39A14, SCG5 and ZFP1 were all upregulated in LSCC and associated with poor prognosis.

There were some limitations in this study. First, although all the known microarray or sequencing data from the public database had been included, the sample size was still not large which may influence the results. Therefore, additional clinical trials with larger

samples may be essential to confirm their expression and prognosis. Second, we only preliminarily predicted that these ceRNA axes may be associated with LSCC development and prognosis. The regulatory relationships between lncRNAs and miRNAs as well as between miRNAs and mRNAs needed further experimental confirmation *in vitro* and *in vivo* (i.e., dual luciferase reporter assay or loss-of-function). Third, whether the expression of TMEM51-AS1 was regulated by methylation should be validated by using the methylation inhibitor 5-azacytidine. Fourth, although we have normalized the data from different platforms, this may still cause some underlying bias.

## CONCLUSION

Our present study identifies several important mechanisms for the development and progression of LSCC: (1) methylation-mediated upregulation of lncRNA TMEM51-AS1 may function as a ceRNA for miR-106b to regulate SNX21 and TRAPPC10; (2) survival-related RUSC1-AS1/LINC00324 may function as a ceRNA to sponge miR-16, miR-10 or miR-7 and then regulate SPRY4/ MICAL2/SLC39A14, SCG5/PRDM5 and ZFP1, respectively. Altogether, these lncRNA, miRNAs or mRNAs may be potential prognostic biomarkers and therapeutic targets of LSCC.

### Funding
This work was supported by Natural Science Foundation of Liaoning Province of PR China (No. 20170541050). The funders had no role in study design, data collection and analysis, decision to publish, or preparation of the manuscript.

### Grant Disclosures
The following grant information was disclosed by the authors:
Natural Science Foundation of Liaoning Province of PR China: 20170541050.

### Competing Interests
The authors declare there are no competing interests.

### Author Contributions

- Lian Hui conceived and designed the experiments, performed the experiments, analyzed the data, contributed reagents/materials/analysis tools, prepared figures and/or tables, authored or reviewed drafts of the paper, approved the final draft.
- Jing Wang performed the experiments, contributed reagents/materials/analysis tools, approved the final draft.
- Jialiang Zhang analyzed the data, contributed reagents/materials/analysis tools, approved the final draft.
- Jin Long conceived and designed the experiments, authored or reviewed drafts of the paper, approved the final draft.

## Data Availability

Raw data is available in the Supplemental Files.

## Supplemental Information

Supplemental information for this article can be found online at http://dx.doi.org/10.7717/peerj.7456#supplemental-information.

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
