# Peer review of "lncRNA TMEM51-AS1 and RUSC1-AS1 function as ceRNAs for induction of laryngeal squamous cell carcinoma and prediction of prognosis"

_PeerJ, doi:10.7717/peerj.7456_

## Round 0.1 · original submission · Major Revisions

Please revise your manuscript according to the reviewers' comments. To make it easier for readers to understandi, please provide the workflows to present your major procedures.

Reviewer 1 ·

Basic reporting

The results are based on bioinformatics analysis and they require to be validated.

Experimental design

no comment

Validity of the findings

no comment

Additional comments

1. The authors have provided a series of bioinformatics analyses among the relationship among expression (mRNA, miRNA, lncRNA), methylation (lncRNA) and clinical outcomes. And it is challenging to evaluate the correctness of their work in a short time. The authors should provide at least a workflow to present their major procedures, especially for the data cleaning steps.

2. Figures 1 and 2 need detailed annotations to present the meanings of them. Figures 4 and 5 might be rearranged according to their findings of ceRNA axes. Figure 6 could be combined with figures 4 and 5 to reinforce the story.

3. Table 3, please state whether the p values were corrected or not.

4. Page 13, line 183-4, what is the test to check whether the expression trend is the same?

5. There are several typos in the manuscript.
p21, line 339, an unexpected "," exists between rare and reports.
p13, the link "(http://127.0.0.1:26738/library/stats/html/wilcox.test.html)" is not correct.

Reviewer 2 ·

Basic reporting

.

Experimental design

.

Validity of the findings

.

Additional comments

In this study Hui and co-workers described re-analysis of the published laryngeal squamous cell carcinoma (LSCC) gene expression data set and methylation profile analyzed by microarray. Here, the authors focus on the interaction relationships between miRNAs and lncRNA/mRNA pairs and their correlation with DNA methylation. They identify a subset of lncRNA-miRNA-mRNA and established them in a ceRNA network. They evaluated the prognostic value of some of the identified biomarkers and also linked them to other clinical features. The author concludes that the lncRNAs may be used for therapeutic and prognostic application in LSCC. This is a nice work, however, sample size seems very low for accurate statistical output. The limitations of the study were also discussed. I have few comments and suggestions to improve this manuscript before the publication is warranted.
1. Abstract: The author did not provide any evidence on therapeutic applications of the lncRNAs in the ceRNA network. Therefore, any conclusion on this issue should be avoided.
2. Materials and methods: How did the author determine the cut-off for differentially expressed lncRNAs and miRNAs? For example |logFC(fold change)| > 0.263 which interprets the real fold change as 1.19, seems not to be a very sensitive threshold to identify robust differentially expressed biomarkers.
3. Materials and methods: Does the P-value reported for the statistical analysis are FDR corrected to avoid the occurrence of false positive?
4. Materials and methods: Please correct the spelling of miRecode to miRcode at line 193.
5. Figure 2: It is not clear what does “contra-regulated” means? I think the figure needs
more explanation to become useful to the readers.
6. Figure 3: The text, nodes, and edges in this figure represents interactions pairs in the
ceRNA network, which I find difficult to visualize. I strongly suggest authors to reproduce a high quality figure and highlight the interaction relationships.
7. Results: To confirm the importance of the OS and RFS DELs, DEMs and DEGs the author investigated the association with other clinical features. Could author also show whether the expression level of the biomarkers is not confounded by other clinical features? In other words, I suggest author to check the prognostic independence of biomarkers against clinical features by multivariate Cox regression?
8. Discussion: needs to be shortened.

---

## Round 0.2 · Minor Revisions

Please improve the manuscript according to the reviewer's final comments.

Reviewer 2 ·

Basic reporting

.

Experimental design

.

Validity of the findings

.

Additional comments

I went through the revised manuscript. The author substantially improved the conceptual background of the manuscript. I am satisfied with their provided point by point response on my comments. However, I still have few comments/suggestions to improve the manuscript.

1. Figure 1. Please correct Set III to Set IV for the methylation profile.

2. Discussion needs to be shortened.

3. In general, written English of the manuscript is poor. I strongly recommend proofreading of the manuscript by a native speaker.

---

## Round 0.3 · accepted · Accept

The manuscript has been improved and is now available for publication in PeerJ.